# Time and Frequency of Social Media Use and Loneliness Among U.S. Adults

**DOI:** 10.3390/ijerph22101510

**Published:** 2025-10-01

**Authors:** Jessica R. Gorman, Hyosin Kim, Kari-Lyn K. Sakuma, Geethika Koneru, Memuna Aslam, Cesar Arredondo Abreu, Brian A. Primack

**Affiliations:** School of Nutrition and Public Health, College of Health, Oregon State University, 2250 SW Jefferson Way, Corvallis, OR 97331, USA

**Keywords:** social media, loneliness, social isolation, internet, adult, mental health

## Abstract

The U.S. loneliness epidemic is associated with substantial morbidity and mortality. While higher social media use (SMU) has been associated with higher loneliness among youth, these associations have not been sufficiently examined in adult populations. Additionally, insufficient research has assessed both SMU time and frequency in the same study. Therefore, the primary aim was to evaluate associations between SMU, both by time and frequency, and loneliness in a nationally representative sample of U.S. adults. We recruited 1512 U.S. adults ages 30–70 in 2023. We assessed loneliness using the NIH PROMIS four-item scale and self-reported SMU time and SMU frequency. Survey-weighted logistic regression models determined associations between both SMU measures and loneliness, controlling for gender, age, sexual orientation, educational attainment, employment status, and marital status. Both SMU time and SMU frequency were independently and linearly associated with loneliness (*p* < 0.001 for both). Although odds of loneliness increased for each increase in frequency, the association between time spent on social media and loneliness demonstrated an inverted U-shape with maximal loneliness in the third quartile of SMU. Results suggest that both time and frequency of SMU may be useful targets for interventions aimed at curbing the negative impact of SMU on loneliness.

## 1. Introduction

Loneliness, which is characterized by the subjective experience of insufficient social connections and resources, is increasingly prevalent worldwide [1]. In the United States, as many as half of adults are lonely [2,3], which is associated with negative health consequences including poor mental health [4,5,6,7], cardiovascular morbidity [8,9], substance use [10], intimate partner violence [11], and increased all-cause mortality [1,12]. Emerging evidence suggests that neurobiological factors [13], cognitive processes [14], and psychosocial factors [15] all contribute to the negative consequences of loneliness.

While social engagement and connection had been declining prior to COVID-19 [16], this trend was amplified by the pandemic [17,18]. A recent advisory from the U.S. Surgeon General estimated the influence of loneliness to be equivalent to smoking 15 cigarettes per day and called for a national strategy to address the epidemic [19,20]. Prior to the COVID-19 pandemic, the U.K. issued similar warnings in association with its appointment of a Minister of Loneliness [21,22].

That report of the Surgeon General identified social media use (SMU) as an important risk factor for loneliness [22]. Studies evaluating associations between SMU and loneliness have been mixed [23,24,25]. While early studies suggested that increased SMU might be associated with positive outcomes such as increased social capital [26], subsequent larger studies that used rigorous clinical outcomes demonstrated negative associations between SMU and loneliness [27]. Although a meta-analysis supported negative associations between SMU and loneliness, it emphasized the urgency of more research among adults, because most prior studies focused on youth. Examination of the association between SMU and loneliness is important among adults because (1) these populations also are heavily exposed to social media, (2) associations demonstrated by studies among youth may or may not translate to adult populations, (3) as digital immigrants, adults may have more difficulty using SM adeptly, and (4) adults represent over 250,000,000 Americans, equating to 75% of the U.S. population.

Another major gap in research knowledge involves the separation of SMU time and SMU frequency, both of which are substantial: depending on the specific population, current data estimate time spent on SMU time to be approximately 2–4 h and frequency to be 100–200 social media “checks” per day [28]. Knowing which of these measures, if either, is more strongly associated with loneliness will assist in the development of nuanced tailoring of public health interventions. A better understanding of these associations will also guide future research.

Thus, we aimed to fill these two major gaps by examining associations among SMU time, SMU frequency, and loneliness in a nationally representative population of U.S. adults ages 30 to 70 years. Our primary aims were to assess SMU time and loneliness (Aim 1) and SMU frequency and loneliness (Aim 2). We also conducted an exploratory analysis including both independent variables in the same model (Aim 3). Although there have been prior mixed results and there is a paucity of data related to adults, we hypothesized that both SMU time (H1) and SMU frequency (H2) would be associated with loneliness.

## 2. Materials and Methods

### 2.1. Participants and Procedures

Participants were recruited with the assistance of Qualtrics Sampling Service [29,30]. Participants were eligible if they were between the ages of 30 and 70 inclusive, able to read English, and able to answer questions using an online interface. Qualtrics used stratified sampling to ensure that each decade (e.g., 30–39, 40–49) had roughly the same number of respondents. Participants received Qualtrics credits redeemable for incentives, and all participants completed informed consent. All data were collected between July and August of 2023, and the study was approved by the Institutional Review Board of Oregon State University (Protocol HE-2023-374).

To ensure data quality, we first conducted a pilot survey with 50 participants. Because the pilot did not reveal difficulties, no changes were made, and those 50 participants were included in the final sample. Of the invited participants, 418 began but did not complete the survey. Qualtrics also removed responses of poor quality due to inconsistencies or uniform responses completed too quickly (n = 531). This could indicate completion by a “bot” or substantial lack of effort, calling data into question. One example of an inconsistent response was when a participant responded in one question that they smoked a pack of cigarettes per day, but in another question, that individual stated that they had smoked 0 cigarettes in their lifetime. The final sample included 1512 participants with a response rate of 61.4%. Individuals who were excluded tended to be younger, male, non-White, and more educated compared to those retained in the study (*p* < 0.01). Weighting adjustments were implemented to correct for nonresponse bias.

### 2.2. Measures

Loneliness. Loneliness was assessed using the Patient-Reported Outcome Measurement Information System (PROMIS) short-form version 4a of the perceived social isolation scale [31]. Participants were asked how strongly they agreed, over the past 7 days, with each of the following four items: “I felt left out,” “I felt that people barely know me,” “I felt isolated from others,” and “I felt that people were around me but not with me.” Participants responded to each item with a 5-point Likert scale. The 4-item scale had strong internal consistency reliability (α = 0.92), which is consistent with other studies [31]. Summing scores for the 4 items, each of which was operationalized as 0–4, yielded a total score between 0 and 16 inclusive. For primary analyses, we dichotomized the item to identify high loneliness using a cutoff score of 10. This threshold was selected based on PROMIS guidance [32]. We selected a priori to dichotomize this variable to improve the interpretability of findings, to make the findings more clinically relevant, and because the loneliness PROMIS scale is neither normally distributed nor appropriate for transformation to a normally distributed variable that would be necessary for linear regression.

Social media use time and frequency. Social media time and frequency were self-reported. All SMU items clearly indicated that participants should report only SMU for personal reasons, not for work. SMU time is known to be different on weekdays and weekends [33]. Therefore, to measure time spent on social media, participants responded using drop-down menus to two questions: (1) “About how many hours a day are you on social media on an average weekday (Monday through Friday)?” and (2) “About how many hours a day are you on social media on an average weekend day (Saturday or Sunday)?” Weekly SMU time was calculated by multiplying weekday time by five and adding twice the weekend time. For the purposes of analysis, these data were collapsed into quartiles. This practice has been used to minimize bias [34,35,36], because individuals systematically over-report social media use [37], relative differences—such as division into quartiles—are more accurate than reported numbers.

To measure SMU frequency, participants responded to drop-down menus for each of the top ten social media platforms at the time of the survey—Facebook, X (formerly Twitter), Reddit, YouTube, LinkedIn, Instagram, TikTok, Snapchat, Pinterest, and WhatsApp—which accounted for the vast majority (>95%) of all SM use according to the Pew Research Center, which is considered the industry standard [38]. For each platform, participants were asked, “On average, about how many times a day do you log on to or check [name of platform].” The average daily frequency of accessing social media was calculated as the sum of the number of times participants logged in across all platforms [39].

Sociodemographic information. Participants self-reported age, gender, sexual orientation, race/ethnicity, educational attainment, employment status, and relationship status.

### 2.3. Statistical Analyses

We first described the data for the complete sample and for those with and without loneliness. We used chi-square (*Χ*^2^) tests to determine bivariate associations between each variable and loneliness. We then built three logistic regression models to examine multivariable associations between SMU and loneliness. To address Aim 1, Model 1 included only SMU time as the independent variable. To address Aim 2, Model 2 included only SMU frequency as the independent variable. Finally, to address exploratory Aim 3, Model 3 included both independent variables in the same model. To test for linear trends, we performed Cochran–Armitage tests [40,41].

Primary models adjusted for all 7 sociodemographic covariates. Primary models also included survey weights that resulted in national representation by age, gender, sexual orientation, race/ethnicity, and educational attainment. To examine the robustness of our results, we performed three planned sets of sensitivity analyses. First, we repeated all models without using survey weights. Second, we repeated analyses with different cutoffs for loneliness (i.e., 9 and 11). Third, we repeated analyses with a parsimonious set of covariates—only those that had a bivariate association with loneliness of *p* < 0.05. Finally, we used a stepped hierarchical regression and compared those results with the primary ones.

Statistical analyses were performed using Stata version 18 (Statacorp, College Station, TX, USA), and we defined statistical significance with a two-tailed α ≤ 0.05.

## 3. Results

Table 1 summarizes the weighted sample. One-quarter (25%) of individuals in the weighted sample were identified as lonely. Respondents reported a median (interquartile range [IQR]) of 21 (9, 42) and checked social media platforms a median of 11 (5, 22) times each day (SE = 0.9). The largest age group was 30–39 years (30%), followed by 40–49 years (25%), 60–70 years (24%), and 50–59 years (21%). Half were women, and 92% identified as heterosexual. Three-fourths (75%) were non-Hispanic White, 15% were non-Hispanic Black, and 11% were from other racial/ethnic categories. Half (49%) of the sample completed high school or less, 33% had attained an associate or bachelor’s degree, and 17% held a master’s or doctorate degree. In bivariate analyses, loneliness was associated with SMU time, SMU frequency, younger age, being non-heterosexual, lower educational attainment, being unemployed, and being unmarried (Table 1).

Table 2 presents the findings of multivariable logistic regression models and Cochrane–Armitage tests. Firstly, Cochrane–Armitage tests confirmed significant linear trends between both SMU patterns (time and frequency as ordinal variables) and loneliness (a binary outcome), with *p*-values < 0.001 for both.

Further inspection through multivariable regressions provided detailed insights into the association between SMU and loneliness. In Model 1, which included the SMU time variable in quartiles as an independent variable, revealed an inverted U-shape with odds of loneliness peaking in quartile 3 (21–41 h of SM time per week), followed by odds of loneliness in quartile 4 (OR = 1.67, 95% CI = 1.09–2.56). Individuals in quartile 3 had twice the odds of loneliness (OR = 2.01, 95% CI = 1.36, 2.97) compared with those who used 0–7 h per week (quartile 1). Thus, H1 was supported.

Model 2 showed that the odds of loneliness increased sequentially with each increase in SMU frequency (Table 2). For example, compared with those who checked social media 0–4 times per day (quartile 1), individuals who checked 12–21 times per day had 1.65 times the odds of loneliness (95% CI = 1.08–2.52), and those who checked ≥ 22 times (quartile 4) had 2.31 times the odds (95% CI = 1.47, 3.62) of loneliness. Thus, H2 was supported.

In Model 3, when both SMU time and SMU frequency were included in the same model, SMU frequency demonstrated a particularly strong association (Model 3 in Table 2 and Figure 1). Individuals in the highest quartile of SMU frequency had nearly twice the odds of reporting loneliness compared to those in the lowest quartile (AOR = 1.90, 95% CI = 1.15, 3.13). Similar to Model 1 results, patterns differed slightly between time and frequency. In particular, associations between SMU time and loneliness retained the inverse U-shape seen in Model 1, although the odds ratios for quartiles 2 (8–20 h) and 4 (42 or more hours) were not statistically significant, most likely due to modest sample sizes. Associations between loneliness and covariates were similar to those in Models 1 and 2 (Table 2).

Because the sensitivity analyses (including unweighted analyses, use of alternative cutoffs for defining loneliness, use of a more parsimonious set of covariates in regression models, and a stepped hierarchical approach) did not change the findings of the main analysis, only the primary findings are included here.

## 4. Discussion

In this sample of U.S. adults ages 30–70, there was a significant association between both measures of SMU (time and frequency) and loneliness, even after adjusting for all measured sociodemographic factors. This remained true whether independent variables were included singly or together in the same model. While SMU frequency showed a clear linear relationship with higher odds of loneliness, SMU time exhibited a slightly inverted U-shaped pattern, with the highest odds of loneliness observed among individuals in the third quartile of SMU time (Q3). Thus, consistent with prior studies focused on youth, our findings also highlight the relevance of SMU-related loneliness among midlife and older adults. To our knowledge, this is the first nationally representative study assessing associations among both time and frequency of SMU and loneliness using a clinically established, validated scale.

Associations between SMU and loneliness depend on various contextual and individual factors [34,42]. In teen and young adult populations, prior research demonstrates associations between SMU and negative psychiatric outcomes likely mediated by negative interactions, social comparisons, and observing missed opportunities for meaningful social experiences [43,44,45]. In older adult populations, however, prior research has been mixed. Some prior research suggests that SMU can be protective by providing opportunities for connection and social support [46]. However, other research among older populations suggests negative associations similar to those seen with youth [47]. Our study may reflect a mixture of these prior research findings, because although we found significant associations, they were not as strong and consistent as the associations seem among youth [34].

The present study design does not allow for definitive conclusions to be drawn about the mechanisms behind why SMU may be differently associated with loneliness in different age groups. A possible reason for weaker associations between SMU and loneliness among older adults is that older adults use social media differently than younger adults. For example, older adults may use social media to stay up to date with close connections as opposed to exploring the lives of and interacting with strangers. It is also possible that, compared with younger individuals, older individuals may engage in selective activities based on interests, hobbies, and clear intentions. Hence, for older individuals, SMU may expand or strengthen existing social networks [48,49,50] and provide unique opportunities for social participation [46,51]. Indeed, a prior study of U.S. adults over 50 found that frequency of social media communication was associated with less loneliness, and that this association was mediated by increased in-person social support and quality of social contact online [52]. However, greater SMU also carries with it risks similar to those seen with youth, such as misunderstandings, negative experiences, and social comparison [43,44,45], as well as the risk of developing an overly involved—and possibly addictive—relationship with SMU [53]. Multiple facets of the overall experience of online social support may also be relevant. It will be useful for future work to more closely examine each of these potential mechanisms.

It is worth considering why there were different patterns noted for each of our key independent variables (SMU time and SMU frequency) in terms of their association with loneliness. As an association with loneliness, frequency may be more impactful than time because heavy frequency combines content issues with the interruption of the normal course of life. Indeed, while the Cochrane–Armitage test found a linear effect, visually, there is a curvilinear effect with a particularly strong increase from Q3 to Q4.

It is particularly interesting to consider why the association between SMU time and loneliness decreased from Q3 to Q4. One possibility is that, compared with those in the third quartile (21–41 SMU hours weekly), individuals in the highest quartile of SMU (42 SMU hours or more weekly) engage in active SMU integral to their primary social networks in addition to using social media for leisure and entertainment. Second, unobserved factors may contribute to this pattern. For example, those reporting very high SMU time may also be more digitally literate, highly socially motivated, and/or intentional in their online interactions. These factors may provide them with relative protection against loneliness compared to those reporting moderately high SMU time. These questions may be best answered by obtaining, in future research, more nuanced quantitative data and/or qualitative data related to these potential mediating factors.

We noted that several sociodemographic factors were associated with loneliness. Those who were single and those who were not working, and therefore may have fewer opportunities for social connection, had higher odds of loneliness. This aligns with the established relationship between social connectedness and loneliness [54]. It is also consistent with other research demonstrating that marital status and employment status can contribute to social isolation [55,56]. Given the observed association between SMU and loneliness, extensive SMU may not be a valuable strategy for alleviating loneliness among individuals already at higher risk due to the status of their relationship, education, or employment. In alignment with prior research, we also noted an association between sexual minority identity and loneliness [57,58,59]. There are likely multiple contextual factors contributing to these relationships that require further in-depth study to identify contributing factors to these associations. Such research may also provide guidance for identifying audiences and settings where interventions would be the most useful.

While it should be emphasized that this is one study limited by its cross-sectional design, it does suggest that considerations around policy and clinical interactions include caution around substantial SMU in midlife and older adulthood—just as it has been suggested for youth and young adults [34].

## 5. Limitations

This was a cross-sectional study; therefore, directionality cannot be determined. For example, it is possible that those who feel socially isolated actively seek out connections on social media. However, it is also possible that those who use social media feel lonelier as a result of experiences arising from SMU. We also relied on SMU self-report, which is subject to recall bias and social desirability bias. For this reason, we operationalized SMU variables in quartiles; while SMU self-report does not reflect exact use, relative amounts tend to be accurate [60]. Another limitation is that we only use a global assessment of time and frequency of SMU instead of nuanced contextual constructs such as reason for use, active vs. passive use, positive vs. negative experiences, and makeup of contacts. Due to small sample sizes in some cells, our results should be interpreted with caution, particularly with regard to the inverse U-shaped curve observed. Lastly, although we applied weights to improve the representativeness of our findings, the number of respondents from certain marginalized populations was limited in the original sampling. Future studies could benefit from intentional oversampling of minority groups to examine this topic within these populations.

## 6. Conclusions

Our study revealed that higher SMU, by both frequency and time, was associated with loneliness in a nationally representative sample of adults. It would be valuable for future studies to examine this association longitudinally and to determine whether these associations hold for different experiences of SMU (e.g., perceived positive vs. negative experiences) and reasons for SMU (e.g., to stay connected to family and friends vs. reading exchanges of celebrities or politicians). Technology-based interventions that support the development of social connections and supportive communication online represent a possible strategy for future intervention to promote healthy SMU habits, reduce loneliness, and improve psychological well-being [52,61]. Based on the state of current research, including the findings of this study, policy makers, public health professionals, and clinicians should remain alert to the potential impacts that social media use can have on loneliness. Those who use social media frequently and for longer durations of time may be most vulnerable. Longitudinal studies, including comprehensive assessments of social interactions online and in person, could elucidate how and why social media use may influence loneliness across age groups.

## Figures and Tables

**Figure 1 ijerph-22-01510-f001:**
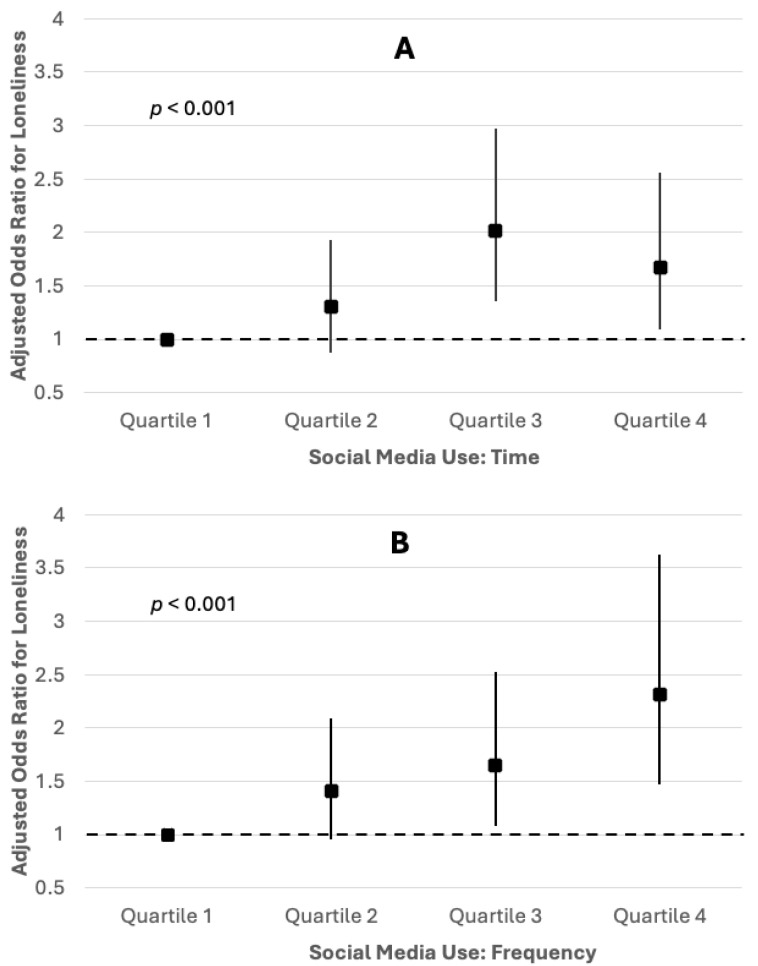
These figures represent odds ratios and 95% confidence intervals for loneliness based on social media use among U.S. adults ages 30–70. Independent variables were time of social media use (**A**) and frequency of social media use (**B**). In terms of time, social media use in hours per week was 0–7 h for quartile 1, 8–20 h for quartile 2, 21–41 h for quartile 3, and 42 or more hours for quartile 4. In terms of frequency, social media use in checks per day was 0–4 h for quartile 1, 5–11 h for quartile 2, 12–21 h for quartile 3, and 22 or more for quartile 4. *p*-values represent linear effect according to the Cochrane–Armitage test.

**Table 1 ijerph-22-01510-t001:** Sample descriptives by loneliness (n = 1512).

Variable	No Loneliness	Loneliness ^1^	*p*-Value for *Χ*^2^	Total n (%)
Actual sample size, n (%)	1120 (74)	392 (26)		1512 (100)
Weighted %	75	25		
SMU time (hours per week), median (IQR)	18 (7, 40)	25 (14, 49)	<0.001	21 (9, 42)
SMU time (hours per week)			<0.001	
0–7	25	17		23
8–20	24	22		23
21–41	23	32		25
42 or more	28	30		28
SMU frequency (checks per day), median (IQR)	10 (4, 20)	13 (6, 27.5)	<0.001	11 (5, 22)
SMU frequency (checks per day, in quartiles)			<0.001	
0–4	25	15		23
5–11	26	25		25
12–21	22	25		23
22 or more	27	35		29
Age			<0.001	
30–39	30	33		30
40–49	23	30		25
50–59	20	25		21
60–70	27	13		24
Gender identity			0.33	
Man	51	48		50
Woman	49	52		50
Sexuality			<0.001	
Heterosexual	93	87		92
Non-heterosexual	7	13		8
Race/ethnicity			0.24	
Non-Hispanic White	75	73		75
Non-Hispanic Black	13	17		14
Other (Hispanic, APAA, multiple race)	12	10		11
Education			0.03	
No degree	47	56		49
Associate’s or bachelor’s degree	35	29		33
Master’s or doctorate degree	18	15		17
Employment			0.01	
Employed	66	62		65
Partially employed ^2^	29	27		28
Unemployed	5	10		7
Marital status			<0.001	
Single	22	35		25
Married	58	44		55
Other ^3^	20	21		20

Abbreviations. APAA: Asian American, Pacific Islander, American Indian, or Alaskan Native. IQR: interquartile range. SMU: social media use. ^1^ Loneliness was defined using the Patient-Reported Outcome Measurement System (PROMIS) Perceived Social Isolation subscale with a cutoff of a T-score of 70, or two standard deviations above the mean. ^2^ Includes part-time or sporadic work. ^3^ Includes divorced, separated, or widowed.

**Table 2 ijerph-22-01510-t002:** Social media use time, frequency, and loneliness among a national sample of 30–70-year-old U.S. adults.

Variable	Model 1		Model 2		Model 3	
	AOR ^a^	(95% CI)	AOR	(95% CI)	AOR	(95% CI)
SMU time (hours per week, in quartiles)						
0–7	1.00				1.00 ^b^	
8–20	1.30	(0.87, 1.93)			1.18	(0.79, 1.76)
21–41	2.01 ***	(1.36, 2.97)			1.62 *	(1.07, 2.44)
42 or more	1.67 *	(1.09, 2.56)			1.23	(0.77, 1.96)
SMU frequency (checks per day, in quartiles)						
0–4			1.00		1.00 ^c^	
5–11			1.41	(0.95, 2.09)	1.21	(0.82, 1.79)
12–21			1.65 *	(1.08, 2.52)	1.35	(0.86, 2.11)
22 or more			2.31 ***	(1.47, 3.62)	1.90 *	(1.15, 3.13)
Age						
30–39	1.00		1.00		1.00	
40–49	1.10	(0.77, 1.56)	1.15	(0.80, 1.64)	1.14	(0.80, 1.62)
50–59	1.04	(0.71, 1.51)	1.14	(0.77, 1.67)	1.14	(0.78, 1.68)
60–70	0.37 ***	(0.22, 0.60)	0.42 **	(0.25, 0.71)	0.42 **	(0.25, 0.71)
Gender identity ^d^						
Man	1.00		1.00		1.00	
Woman	1.16	(0.88, 1.52)	1.16	(0.88, 1.53)	1.16	(0.88, 1.53)
Sexuality						
Heterosexual	1.00		1.00		1.00	
Not heterosexual	1.63 *	(1.06, 2.53)	1.66 *	(1.07, 2.57)	1.64 *	(1.06, 2.55)
Race						
Non-Hispanic White	1.00		1.00		1.00	
Non-Hispanic Black	0.94	(0.64, 1.39)	0.92	(0.62, 1.36)	0.90	(0.61, 1.33)
Other (Hispanic, APAA, multiple race)	0.75	(0.49, 1.17)	0.74	(0.48, 1.16)	0.74	(0.47, 1.15)
Education						
No degree	1.00		1.00		1.00	
Associate’s or bachelor’s degree	0.82	(0.61, 1.10)	0.78	(0.58, 1.04)	0.79	(0.59, 1.06)
Master’s or doctorate degree	0.89	(0.57, 1.38)	0.80	(0.52, 1.25)	0.82	(0.53, 1.28)
Employment						
Employed	1.00		1.00		1.00	
Partially employed ^e^	1.51 *	(1.08, 2.13)	1.60 **	(1.13, 2.25)	1.58 **	(1.12, 2.24)
Unemployed	1.81 *	(1.13, 2.91)	1.86 *	(1.16, 2.99)	1.89 **	(1.18, 3.03)
Marital status						
Single	1.00		1.00		1.00	
Married	0.54 ***	(0.40, 0.74)	0.54 ***	(0.40, 0.75)	0.53 ***	(0.39, 0.73)
Other ^f^	0.83	(0.56, 1.22)	0.82	(0.56, 1.21)	0.82	(0.56, 1.21)

Abbreviations. SMU: social media use. AOR: Adjusted Odds Ratio. APAA: Asian American, Pacific Islander, American Indian, or Alaskan Native. CI: confidence interval. Levels of Statistical Significance. *** = *p* < 0.001, ** = *p* < 0.01, * = *p* < 0.05. ^a^ Odds ratio, adjusted for all variables in the table, for loneliness as defined using the Patient-Reported Outcome Measurement System (PROMIS) Perceived Social Isolation subscale with a cutoff at a T-score of 70, equivalent to two standard deviations above the mean. ^b^
*p* < 0.001 for linear trend from Cochrane–Armitage test (time of SMU vs. loneliness). ^c^
*p* < 0.001 for linear trend from Cochrane–Armitage test (frequency of SMU vs. loneliness). ^d^ The 9 individuals who indicated a gender identity other than man or woman were not included in multivariable analyses to avoid instability of models resulting from small cell sizes. ^e^ Includes part-time or sporadic work. ^f^ Includes divorced, separated, or widowed.

## Data Availability

The data that support the findings of this study are available from the corresponding author upon reasonable request.

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
