# Peer review of "Time and Frequency of Social Media Use and Loneliness Among U.S. Adults"

_ijerph, 2025, doi:10.3390/ijerph22101510_

Round 1
Reviewer 1 Report
Comments and Suggestions for Authors
The article deals with a very topical issue: the relationship between loneliness and social media use (SMU) in a nationally representative sample of adults in the US. Since this connection has primarily been studied in adolescents to date, the authors focus on adults between the ages of 30 and 70. Because studies often focus on either the duration or frequency of social media use, this study examined both aspects of SMU in relation to loneliness within a single model.
The manuscript is of high quality, although it is another cross-sectional study that relies on self-reports with regard to SMU. However, given the strengths of the study mentioned above and the authors' awareness of its weaknesses (see Limitations section), I would suggest accepting the study subject to minor revisions.
There are a few aspects that could be considered to improve the quality of the manuscript:
Measures:
- It should be clarified which platforms were considered social media. It appears that you also included messengers such as WhatsApp. Was this clearly explained to the participants? Could you please explain why you decided to include messengers in your analyses? I think it might be helpful to repeat the analysis to check whether there are differences in loneliness depending on the type of social media (only “classic” social media vs. only messengers vs. all)
Discussion:
- Please also discuss the presumed mechanism, why a SMU time of 42 hours or more could be associated with lower loneliness levels than SMU time of 21-41 hours.
- In lines 233 to 240, you examine the relationship between SMU, loneliness, and age. You should emphasize that your study does not allow conclusions to be drawn about the mechanisms behind why SMU may be more or less beneficial in certain age groups. In addition, you should critically reconsider your conclusion (shift to a conscious SMU for younger adults and frequent social media users in favour of increased in-person social support and quality of social contact online) in the context of online social support, as this carries the risk of developing a close emotional bond to social media, which can lead to addictive SMU (cf. e.g., Brailovskaia, 2024). Please mention that the experience of online social support can also be seen as a risk factor.
Limitations:
- Please also note the small sample sizes in the individual cells, which have influenced the results and should therefore be interpreted with caution (especially the inverted U-shape of the correlation between loneliness and SMU time).
Conclusion:
- Practical implications for policymakers and clinicians are lacking and should be better elaborated.
List of References:
- Some references are incorrectly cited (e.g., No. 24 with full authors names). Please check the complete list for any errors.
Author Response
Reviewer 1 comments:
The article deals with a very topical issue: the relationship between loneliness and social media use (SMU) in a nationally representative sample of adults in the US. Since this connection has primarily been studied in adolescents to date, the authors focus on adults between the ages of 30 and 70. Because studies often focus on either the duration or frequency of social media use, this study examined both aspects of SMU in relation to loneliness within a single model.
The manuscript is of high quality, although it is another cross-sectional study that relies on self-reports with regard to SMU. However, given the strengths of the study mentioned above and the authors' awareness of its weaknesses (see Limitations section), I would suggest accepting the study subject to minor revisions.
There are a few aspects that could be considered to improve the quality of the manuscript:
Measures:
- It should be clarified which platforms were considered social media. It appears that you also included messengers such as WhatsApp. Was this clearly explained to the participants? Could you please explain why you decided to include messengers in your analyses? I think it might be helpful to repeat the analysis to check whether there are differences in loneliness depending on the type of social media (only “classic” social media vs. only messengers vs. all)
In the revision, we clarify that we selected the top ten commonly used social media platforms in the United States as described by the Pew Research center—the industry standard—at the time of the study (lines 123-126, reference 38). We presented these to the participants so they would know which platforms were considered social media when they responded to questions. WhatsApp is included in this list as a social media platform (line 123)
Discussion:
- Please also discuss the presumed mechanism, why a SMU time of 42 hours or more could be associated with lower loneliness levels than SMU time of 21-41 hours.
In the revision, we discuss two possible mechanisms (lines 269-278). First, it may be that individuals in the highest quartile of SMU engage in active SMU integral to their primary social networks in addition to using social media for leisure and entertainment. Second, unobserved factors may contribute to this pattern. For example, those reporting very high SMU time may be more digitally literate, highly socially motivated, and/or intentional in their online interactions. This may provide them with relative protection against loneliness compared to those reporting moderately high SMU time.
- In lines 233 to 240, you examine the relationship between SMU, loneliness, and age. You should emphasize that your study does not allow conclusions to be drawn about the mechanisms behind why SMU may be more or less beneficial in certain age groups. In addition, you should critically reconsider your conclusion (shift to a conscious SMU for younger adults and frequent social media users in favour of increased in-person social support and quality of social contact online) in the context of online social support, as this carries the risk of developing a close emotional bond to social media, which can lead to addictive SMU (cf. e.g., Brailovskaia, 2024). Please mention that the experience of online social support can also be seen as a risk factor.
In the revision we made four responses in response to this comment (lines 234-261). First, we emphasize clearly that our study does not allow definitive conclusions to be drawn about mechanisms behind why SMU may be more or less beneficial among different age groups. Second, we clarified that our prior conclusion is only one possible mechanism and that there are other possible explanations to consider. Third, we add the consideration the reviewer suggests related to addictive SMU, including a reference to the Brailovskskaia research mentioned (reference 53). Fourth, we mention explicitly that, as the reviewer indicates, the experience of online social support could be a risk factor.
Limitations:
- Please also note the small sample sizes in the individual cells, which have influenced the results and should therefore be interpreted with caution (especially the inverted U-shape of the correlation between loneliness and SMU time).
In the revision, we add this consideration related to small sample sizes in individual cells (lines 305-307).
Conclusion:
- Practical implications for policymakers and clinicians are lacking and should be better elaborated.
In the revision we elaborate on implications for both policymakers and clinicians (lines 321-325).
List of References:
- Some references are incorrectly cited (e.g., No. 24 with full authors names). Please check the complete list for any others.
In the revision, we present a cleaner set of references that fixes the specific error mentioned. We also carefully examined other references for similar mistakes and corrected them.

Reviewer 2 Report
Comments and Suggestions for Authors
Study looks at social media use and frequency in American adults aged over 30. Important and well thought out study introduction covers topic well but would benefit from more clearly defined hypotheses. Methods and results are mostly clear, some details on any check questions used should be included. Dichotomising loneliness as a variable requires a little more justification for why this was done, dichotomising a continuous scale in this way is problematic because it reduces statistical power, obscures individual differences, and imposes an arbitrary cut-off that may not accurately reflect meaningful distinctions in loneliness. Equally quartering social media time involves the same issue, here the minimum authors should provide the ranges and quartile points at a minimum. (In which case the categorisation should be described). A stepped hierarchical regression may be more appropriate than running multiple separate models, since the latter approach essentially explains the same variance in the data using different subsets of predictors. Using a hierarchical approach would also allow you to enter demographic factors in the first step to account for any variance they explain before adding your main predictors. To aid interpretation, a visual representation of the findings from model three should be included. In addition, it would be valuable to examine usage patterns of individual social media platforms, and to explore whether any specific platforms show associations with demographics or loneliness. The discussion section is somewhat brief but well considered. Expanding on the implications and potential future research directions would strengthen the overall contribution of the paper and provide greater context for the findings.
Author Response
Reviewer 2 comments:
Introduction:
- Study looks at social media use and frequency in American adults aged over 30. Important and well thought out study introduction covers topic well but would benefit from more clearly defined hypotheses.
In the revision, we present the clearly defined hypotheses that that, consistent with prior research among youth, higher SMU time (H1) and SMU frequency (H2) would be associated with a greater odds of loneliness (lines 68-70). Inclusion of these specific hypotheses also necessitated language in the results section stating that both hypotheses were supported (Results, lines 178 and 183).
Materials and Methods:
- Methods and results are mostly clear, some details on any check questions used should be included.
In the revision, we add the precise wording from survey questions when needed (lines 112-114 and 125-126).
- Dichotomising loneliness as a variable requires a little more justification for why this was done, dichotomising a continuous scale in this way is problematic because it reduces statistical power, obscures individual differences, and imposes an arbitrary cut-off that may not accurately reflect meaningful distinctions in loneliness.
In the revision, we make three changes in response to this comment. First, we present a strengthened justification for using loneliness as a binary outcome (lines 104-107). In particular, using this particular loneliness scale as continuous is not statistically recommended because of its distribution. Second, we present sensitivity analyses using various cutoff points to test the robustness of results (lines 143-144).
- Equally quartering social media time involves the same issue, here the minimum authors should provide the ranges and quartile points at a minimum (In which case the categorisation should be described).
In the revision, we clarify that quartiles are most ideal for the independent variable in this case (lines 117-119). Second, we clearly provide the ranges and quartile points as the reviewer requests (Tables 1 and 2).
- A stepped hierarchical regression may be more appropriate than running multiple separate models, since the latter approach essentially explains the same variance in the data using different subsets of predictors. Using a hierarchical approach would also allow you to enter demographic factors in the first step to account for any variance they explain before adding your main predictors.
In the revision, we include a sensitivity analysis that included a stepped hierarchical model (lines 145-146). Because the results were similar to those of our a priori primary analyses, we present only the primary results in the manuscript (lines 195-196).
Results:
- To aid interpretation, a visual representation of the findings from model three should be included. In addition, it would be valuable to examine usage patterns of individual social media platforms, and to explore whether any specific platforms show associations with demographics or loneliness.
In the revision, as suggested by the reviewer, we include a visual representation of the key results from model 3 (Figure).
Discussion:
- The discussion section is somewhat brief but well considered. Expanding on the implications and potential future research directions would strengthen the overall contribution of the paper and provide greater context for the findings.
In response to this, as well as a similar comment from reviewer 1, in the revised discussion we expand on the implications of the results and potential future research directions (lines 236-261, 269-278, 292-295, 321-325).
